# Overexpression of *Liriodenron WOX5* in *Arabidopsis* Leads to Ectopic Flower Formation and Altered Root Morphology

**DOI:** 10.3390/ijms24020906

**Published:** 2023-01-04

**Authors:** Dandan Wang, Xiaoxiao Ma, Zhaodong Hao, Xiaofei Long, Jisen Shi, Jinhui Chen

**Affiliations:** Key Laboratory of Forest Genetics & Biotechnology of Ministry of Education, Co-Innovation Center for Sustainable Forestry in Southern China, Nanjing Forestry University, Nanjing 210037, China

**Keywords:** *LhWOX5*, ectopic flower formation, altered root morphology, *Liriodendron* hybrid

## Abstract

Roots are essential for plant growth, and studies on root-related genes, exemplified by *WUSCHEL-RELATED HOMEOBOX5* (*WOX5*), have mainly concentrated on model organisms with less emphasis on the function of these genes in woody plants. Here, we report that overexpression of the *WOX5* gene from *Liriodendron* hybrid (*LhWOX5*) in *Arabidopsis* leads to significant morphological changes in both the aerial and subterranean organs. In the *Arabidopsis* aerial parts, overexpression of *LhWOX5* results in the production of ectopic floral meristems and leaves, possibly via the ectopic activation of *CLV3* and *LFY*. In addition, in the *Arabidopsis* root, overexpression of *LhWOX5* alters root apical meristem morphology, leading to a curled and shortened primary root. Importantly, these abnormal phenotypes in the aerial and subterranean organs caused by constitutive ectopic expression of *LhWOX5* mimic the observed phenotypes when overexpressing *AtWUS* and *AtWOX5* in *Arabidopsis*, respectively. Taken together, we propose that the *LhWOX*5 gene, originating from the Magnoliaceae plant *Liriodendron*, is a functional homolog of the *AtWUS* gene from *Arabidopsis*, while showing the highest degree of sequence similarity with its ortholog, *AtWOX5*. Our study provides insight into the potential role of *LhWOX5* in the development of both the shoot and root.

## 1. Introduction

The WUSCHEL HOMEOBOX (WOX) gene family is found only in plants and plays an important role in plant development and growth [1,2]. Two signaling loops critical for plant development are the *WUSCHEL/CLAVATA3* (*WUS/CLV3*) and *WUSCHELRELATED HOMEOBOX 5*/*CLAVATA3/ESR-RELATED40* (*WOX5/CLE40*) pathways, which both have a member of the WOX gene family at their center and are known for their important roles in shoot and root apical meristem maintenance, respectively [3,4]. In addition, *WUS* is involved in the maintenance of floral meristems, together with a collection of other developmental genes such as *AGAMOUS* (*AG*), *KNUCKLES* (*KNU*), *CLV3,* and *LEAFY* (*LFY*), which regulate the complete flowering pathway [5,6]. Of the genes mentioned above, *LFY* is involved in the first step of the transition from vegetative to reproductive development [7]. To be more specific, *LFY* induces floral meristem development and controls their morphology by activating floral organ identity genes such as *AG* [5,8].

It has been demonstrated that *AtWUS* can substitute for *WOX5* when expressed in the quiescent center (QC) of the root apical meristem and vice versa, but they are interchangeable only in stem cell maintenance [9]. *WOX5/7* of *Arabidopsis* is a pair of genes with a high degree of homology [2]. *WOX5* is involved not only in the development of primary and lateral roots but also in leaf medio-lateral axis formation [10]. Whereas *WOX7* was found to be involved in the development of lateral roots in response to sugar, the number of lateral root primordia was increased in a *wox7* mutant but reduced upon over-expressing *WOX7* [11]. 

Overexpression of *AtWOX5* resulted in the absence of starch granules, which are involved in gravitropic sensing, in the differentiated column cell zone, whereas in the *wox5* mutant, column stem cells become differentiated and contain starch granules [9]. It was subsequently confirmed that the WOX5-IAA17 (indole-3-acetic acid 17) feedback circuit mediates the maintenance of the auxin gradient in the root tip, which is crucial for the patterning of root stem cell niches in *Arabidopsis*. [12]. Furthermore, WOX5 functions together with TOPLESS/TOPLESS-RELATED (TPL/TPR) co-repressors and HISTONE DEACETYLASE 19 (HDA19) to silence the differentiation factor *CYCLING DOF FACTOR 4* (*CDF4*) and regulate stem cell maintenance [13].

Heterologous expression of *Arabidopsis WOX5/WUS* in tobacco can significantly alter the morphology of the tobacco root tip and induce adventitious shoot formation within the root region, which is similar to the phenotype of *WUS* overexpressing tobacco itself [14,15,16]. Because of the importance of the *WOX5* gene in root tip development in *Arabidopsis*, many functional studies of *WOX5* in other species have also been reported. For example, *QUIESCENT-CENTER-SPECIFIC HOMEOBOX* (*QHB*), a *WOX5* ortholog in rice, is expressed in the QC and metaxylem, while a similar CLE-WOX regulatory loop as is present in *Arabidopsis* functions in the root apical meristem in rice [17]. Over-expressed *PtWOX5* in *Populus trichocarpa* promotes lateral root organogenesis but inhibits lateral root growth by restricting cell division and suppressing differentiation-related genes such as D-type cyclins (*CYCD*) [18]. Interestingly, overexpression of wheat *TaWOX5* can reduce genotype dependence during wheat gene transformation while improving the efficiency of genetic transformation and genome editing in wheat and other crops, which is an important aspect in the application of the *WOX5* gene [19].

There are many reports on *WOX5* function in various crops, as mentioned above, but limited research has been performed on the *WOX5* ortholog in woody plants, especially in non-model organisms. *Liriodendron* is widely distributed in Eastern North America [20] and has a high commercial value in China [21]. *Liriodendron* belongs to the Magnoliidae family, which takes up an evolutionarily intermediate position in between basal angiosperms and eudicots [20]. The study of *Liriodendron* may help us better understand the functional diversification of genes during plant evolution. Moreover, *Liriodendron* has a mature somatic embryogenesis and transgenesis system [21], making it an ideal woody plant. Although transgenic *Liriodendron* can be obtained through gene transformation, it requires significantly more time than the quintessential model plant, *Arabidopsis*. 

Here, we performed functional studies on *LhWOX5* by expressing it heterologously in *Arabidopsis*, comparing a *WOX5* gene originating from magnoliids to its eudicot counterpart. We provide evidence that *LhWOX5* shares functional homology not only with *AtWOX5* but also with *AtWUS*, suggesting that both *Arabidopsis* genes may have diverged from an ancestral gene that combined both of their functionalities. Hypothetically, the dramatic phenotype caused by *LhWOX5* overexpression, being the ectopic induction of floral organs, could be applied in an industrial setting to stimulate flower production in i.e., ornamental plants.

## 2. Results

### 2.1. Identification and Tissue-Specific Expression Pattern Analysis of LhWOX5

To identify the WOX5 ortholog protein in *Liriodendron*, we searched the genome [20] and transcriptome of *Liriodendron* using the WOX5/7 protein sequences from *Arabidopsis thaliana*, *Vitis vinifera*, and *Oryza sativa* as queries. We identified a putative *WOX5* gene and, through protein sequence alignment, found that this gene contains the typical conserved HD domain and shares the same WUS-box (amino acids, TLLFP) with the WOX5 genes from rice, *Arabidopsis,* and grape [1,2] (Figure 1A). Furthermore, a phylogenetic tree showed that this candidate gene from *Liriodendron* belongs to the WOX5 clade (Figure 1B). Therefore, we named this protein LhWOX5. 

To examine the expression pattern of *LhWOX5* in *Liriodendron* hybrid seedlings, we collected several major tissues for quantitative reverse transcription PCR (qRT-PCR) analysis. We found that *LhWOX5* is expressed most strongly in the bud, followed by the root and stalk, with the lowest expression observed in the leaf (Figure 2A). This suggests that *LhWOX5* may function in root and shoot development.

### 2.2. LhWOX5 Overexpression Causes Ectopic Organogenesis in the Arabidopsis Rosette, Leaf, and Stalk

To further investigate the function of *LhWOX5*, we overexpressed this gene in *Arabidopsis* using the 35S cauliflower mosaic virus promoter. More than 50% of the T1 generation plants displayed flower buds and new leaves on the veins of rosette leaves and a curly stalk with undetermined hyperplasia (Figure 3A–D). From the T2 generation, we counted 102 plants from three individual lines (T2-1, T2-2, and T2-3) and found that 45% of the plants showed incomplete cotyledons, 53% showed curled stalks, 45% showed rosette leaves with ectopic leaves and flower buds, and 21% of the plants showed fused stalks (Figure 3E–H). It shows that the transgene-induced phenotype can be passed down from the T1 to the T2 generation.

To investigate whether shoot architecture had been altered or not, we counted the number of branches, inflorescences, and rosette leaves, then determined plant height for the overexpression lines (Table 1). The number of leaves and plant height were not significantly different in the overexpression lines in comparison to the control lines, and only a single overexpression line showed an increased number of branches and inflorescences, respectively (Table 1). Taken together, these results show that overexpression of *LhWOX5* not only promotes new organs on the abaxial side of leaves but also slightly affects the number of branches and inflorescences in different lines.

### 2.3. LhWOX5 Overexpression Leads to the Transcriptional Upregulation of LFY and CLV3

We wondered what specific genes might be responsible for ectopic flowers and leaves induced by *LhWOX5* overexpression. We focused on the *CLV3* and *LFY* genes because of their importance in floral and shoot meristem identity, respectively. *LFY* participates in the formation of floral meristem identity and is strongly expressed in floral primordia [22,23], while *CLV3*, a marker for the shoot meristem, forms a regulatory loop with *WUS* for shoot apical meristem maintenance [24]. We collected tissues as shown in Figure 4B–G and analyzed *LFY* and *CLV3* expression levels via qRT-PCR. We found that the expression of *LFY* and *CLV3* was significantly increased (Figure 4A), suggesting that the formation of ectopic organs on leaf veins is likely due to the abnormal expression of *LFY* and *CLV3*, which was induced by *LhWOX5*.

### 2.4. LhWOX5 Overexpression Affects Arabidopsis Root Morphology and Root Length

It is well known that the *WOX5* gene is associated with root development, and our preliminary data shows that *LhWOX5* is indeed expressed in roots (Figure 2A). We therefore asked whether overexpression of *LhWOX5* affects root development and morphology. Thus, we examined the phenotype of the roots using *LhWOX5* overexpressing plants of the T3-generation. The overexpression lines showed 30–60% curly roots (Figure 5A–D, Table 2) in 8-day seedlings, suggesting changes in root morphology. Meanwhile, 4.70% (12/254) of the overexpression lines have a flower-like shoot structure growing from their root (Appendix A). We also found that the root length was significantly reduced in all three transgenic lines (Figure 5E). We observed 5–7 days root tips and found that the cells of overexpression root tips are not as well layered as in the wild type (Figure 6). In overexpressing lines, we found that there are more undifferentiated cells and fewer starch grains present in the root apical areas, which might account for the observed root tip curling Figure 6 and Appendix A).

## 3. Discussion

Plant roots are fundamental for plant growth, and their morphology strongly influences their ability to absorb nutrients and water. A stable root system can ensure that woody plants grow to form wood or produce fruit. Several factors have been found to affect root growth in the model organism *Arabidopsis thaliana* [25]. *WOX5*, a member of the WOX family, has been intensively studied because of its important role in root stem cell niche maintenance. In woody plants, Li et al. found that *PtWOX5* in poplar promotes lateral root initiation but inhibits lateral root growth [18]. However, similar studies in woody plants are relatively rare due to their long life cycles and the fact that transgenesis is still uncommon. Here, we identified a single copy of the *WOX5* gene in a *Liriodendron* hybrid and performed a pilot study on the function of this gene by overexpressing it in *Arabidopsis* through a heterologous expression system. 

Although previous studies have demonstrated that *WOX5* has an important role in root development in many different plant species [9,14,18,19,26,27], we found that the expression pattern of *LhWOX5* is not the highest in the root of *Liriodendron* hybrid. The highest *LhWOX5* expression is present in the bud, suggesting that *WOX5* may not only affect root development but also shoot apical formation in this species. 

Overexpression of *LhWOX5* in *Arabidopsis* activates ectopic leaf and floral formation from the rosette leaves, as well as ectopic cell proliferation at curled stalk sites (Figure 3), a phenotype that is similar to that of *AtWUS* overexpression in *Arabidopsis*, which can induce ectopic flower buds on non-reproductive organs [28], suggesting that *LhWOX5* may function similarly to *AtWUS* in converting nutritive organs to reproductive organs in *Arabidopsis*.

Overexpression of *LhWOX5* in *Arabidopsis* promotes the expression of a set of meristem-related genes, such as *AtCLV3* and *AtLFY*. Notably, we found *AtLFY* to be expressed at an exceptionally high level, which might be related to the fact that we collected materials with numerous ectopic flower buds rather than ectopic leaves. Furthermore, overexpression of *AtWUS* in *Arabidopsis* elevated expression of *AtCLV3*, *AtLFY*, and *AtAG* [28]. *AtLFY* promotes the transition from vegetative to reproductive organs and the differentiation of floral meristems [22,29]. It is expressed at subsequent stages of floral development, which may explain the production of ectopic floral buds in both *AtWUS* and *LhWOX5* overexpressed lines [28]. In fact, when *AtLFY* is overexpressed in *Arabidopsis*, ectopic flowers develop at the base of rosette leaves [23]. These data suggest that *LhWOX5* may share functional homology with *AtWUS* in initiating floral meristems on non-reproductive organs in *Arabidopsis* and that they may accomplish this process via inducing *AtLFY* expression.

We found root length to be significantly shorter in all *OE-LhWOX5* lines compared to WT (Figure 5E). This could be explained due to stem cell production being increased in the root tip, causing less differentiation to occur and the root to grow abnormally (Figure 6 and Appendix A), as it does upon *AtWOX5* overexpression in *Arabidopsis* (Appendix A) [25]. The morphology of the root tip was altered, with a reduced number of starch grains (Figure 6 and Appendix A), a defect that may lead to loss of gravitropic sensing and a curly root phenotype, a phenotype that is also similar to that observed upon *AtWOX5* overexpression in *Arabidopsis* [9,25]. Moreover, *OE*-*AtWOX5* induced nutrient leaf growth in rosette leaves [25], analogous to the *OE*-*LhWOX5* phenotype (Figure 3). These findings suggest functional similarity between *AtWOX5* and *LhWOX5*.

Thus, heterologous expression of the *LhWOX5* gene in *Arabidopsis* causes very similar phenotypes to those of overexpressed *AtWUS* and *AtWOX5*, implying a biochemical functional similarity. We propose that the LhWOX5 protein in *Liriodendron* may combine the functions of both the AtWUS and AtWOX5 proteins from *Arabidopsis,* as it affects the development of the root apical meristem and the formation of floral meristems equally. 

We found *LhWOX5* overexpression to have a dramatic effect on plant architecture, leading to the overproduction of floral organs as well as the alteration of shoot and root morphology. Potentially, being able to increase the floral organ production can be very useful for certain industrial applications of plants, for example in the ornamental sector or in fruit crops/trees. Furthermore, members of the WOX gene family have been shown to contribute to a plant’s ability to cope with stress via direct control of root growth, such as in poplar, where *PagWOX11/12a* stimulates root elongation and biomass growth, helping it to resist drought [30]. Therefore, *LhWOX5* may prove to have industrial applicability in the plant biotechnology sector in the future.

## 4. Materials and Methods

### 4.1. Plant Materials and Growth Conditions

*Arabidopsis thaliana* Columbia ecotype-0 (Col-0) and *J2341* were provided by Prof. Thomas Laux (Signalling Research Centres BIOSS and CIBSS, Faculty of Biology, University of Freiburg, Germany). Wild type (Col-0), T3 generation, *J2341*, *p35S: AtWOX5* in *J2341* seeds were cultured on 1/2 MS medium without antibiotic selection (kanamycin), while three overexpression (T1/T2 generation) lines were cultured on 1/2 MS medium containing 50 mg/L kanamycin. After 2 days of cold stratification, all seeds were germinated in an incubator. Plants were transplanted onto soil after the first pair of true leaves appeared and placed in an incubator. Growth conditions were 22 °C with a 16 h light/8 h dark cycle and 70% humidity [31].

### 4.2. Identification of the LhWOX5 Gene

To identify the *LhWOX5* gene, we retrieved sequences of WOX5 and its close homolog WOX7 from *Arabidopsis thaliana*, *Amborella trichopoda*, *Vitis vinifera*, and *Oryza sativa* (http://planttfdb.gao-lab.org/index.php) and compared these WOX5 sequences with the genome [20] and transcriptome (unpublished) of *Liriodendron* by a local blast to find candidate genes. The candidate sequences were then aligned with the WOX5 amino acid sequences from the species mentioned above by MAFFT [32]. Texshade [33] was used to visualize conserved domains. Multiple full-length sequences (the candidate sequences with all WOXs from *Arabidopsis thaliana*, *Amborella trichopoda, Selaginella moellendorffii*, *Physcomitrella patens subsp. patens*, *Populus trichocarpa*, and *Vitis vinifera* (http://planttfdb.gao-lab.org/index.php) (accessed on 10 October 2022) were aligned using MAFFT [32]. RAxML v8.2.11 [34] was used to construct a phylogenetic tree with the PROTGAMMAAUTO mode and 1000 bootstrap replications to determine to which branch the candidate sequences belonged. 

### 4.3. qRT-PCR Analysis

Material derived from four *Liriodendron* hybrid seedling tissues (leaf, shoot apex (1 cm), shoot (1 cm), and root (1 cm); Figure 3) was collected for the analysis of the *LhWOX5* gene expression pattern. Total RNA was obtained using a Bioteke plant total RNA extraction kit (RP3301). Quantitative real-time PCR (qRT-PCR) reactions were performed using the Vazyme AceQ qPCR SYBR Green Master Mix (without ROX) (Q121-02) on a LightCycler 480 II (Roche). For each sample, three technical and biological replicates were used, and the result was normalized with 18S rRNA as a reference. 

The same qRT-PCR protocol was applied for expression analysis in *LhWOX5* overexpressing lines. Rosette leaves with flower buds and leaves and curly stems with undetermined hyperplasia were collected, using *UBQ10* as an internal reference. The primers used for qRT-PCR are listed in Appendix A. Expression data were calculated using the Livak calculation method [35] and visualized by GraphPad Prism 8 (https://www.graphpad-prism.cn/) (accessed on 25 Octorber 2022).

### 4.4. Gene Cloning, Transformation, and Screening of Transgenic Plants

Based on the *LhWOX5* gene sequence we obtained in the previous step, we designed primers (LhW5F: GAAGATCCGATCATAGAAACAGAG and LhW5R: CTCAGATTGGAATCGTATCCG) and extracted RNA from a *Liriodendron* hybrid leaf from the campus of Nanjing Forestry University (Nanjing, China). *LhWOX5* cDNA was then cloned into pBI121 to generate the *p35S:LhWOX5* vector. The vector was transformed into the *Agrobacterium* strain GV3101 and subsequently into wild type *Arabidopsis* (Col-0) using the floral dip transformation method [31]. Transgenic plants were grown on kanamycin-containing selection medium, then PCR tested using specific primers (35S-F: TGAAGATAGTGGAAAAGGAAGGTG and LhW5R: CTCAGATTGGAATCGTATCCG) to confirm the presence of the transgene.

### 4.5. Data Analysis and Microscopy

Statistical analysis was performed using IBM SPSS Statistics (26) and bar charts were visualized with GraphPad Prism 8. Microscopy analysis of 5–7 days root was performed using 10% glycerin/10 mg/mL propidium iodide (PI) [13] as mounting solution with a Zeiss LSM 800 system. Starch granules in the 5–7 days root apical region were visualized with 1% lugol solution [36] on a Zeiss Axio Vert A1 microscope.

## 5. Conclusions

In summary, we identified and cloned the *LhWOX5* gene in *Liriodendron* and overexpressed it in *Arabidopsis*, identifying strong phenotypes in both the aerial and subterranean parts of transgenic plants. First, in the aerial parts, the overexpression of *LhWOX5* induced ectopic floral meristems and even the formation of mature flowers, while in the root, it caused an altered root apical cell morphology, resulting in curly roots. These phenotypes mimic those of overexpressed *Arabidopsis AtWUS* and *AtWOX5*, respectively. We propose that *LhWOX5* from *Liriodendron* may combine the functionality of *Arabidopsis AtWUS* and *AtWOX5*. Such findings suggest that the WOX genes in woody plants may have a more complex functionality than their herbaceous plant homologues and provide a basis for us to investigate the function of *LhWOX5* in *Liriodendron*. Furthermore, we speculate that heterologous expression of this gene may prove to be a useful application in the ornamental plant and fruit tree cultivation sectors in the future. 

## Figures and Tables

**Figure 1 ijms-24-00906-f001:**
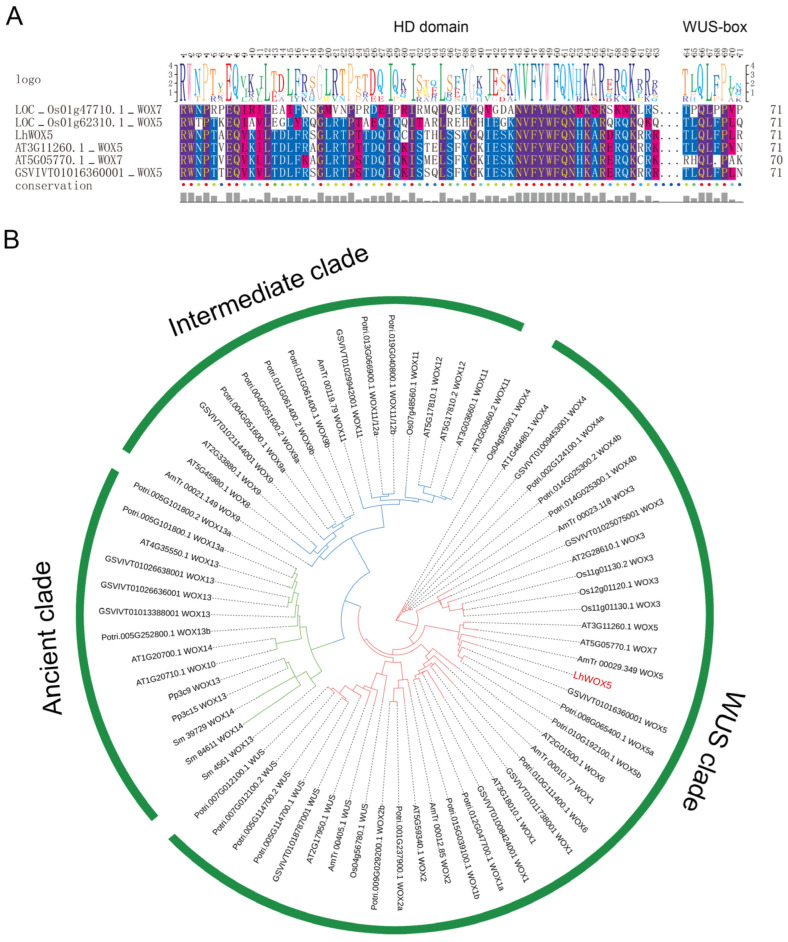
The identification of LhWOX5 in *Liriodendron* hybrids. (**A**) Protein sequences from *Arabidopsis thaliana* (AT3G11260.1_WOX5 and AT5G05770.1_WOX7), *Amborella trichopoda* (AmTr_00029.349_WOX5), *Vitis vinifera* (GSVIVT01016360001_WOX5), and *Oryza sativa* (LOC_Os01g62310.1_WOX5 and LOC_Os01g47710.1_WOX7) were aligned. The conserved Homeodomain (HD domain) [1] and WUS-box [2] are shown. (**B**) Phylogeny of the WOX gene family from various species. WOX genes from *Arabidopsis thaliana*, *Amborella trichopoda, Selaginella moellendorffii*, *Physcomitrella patens subsp. patens*, *Populus trichocarpa*, and *Vitis vinifera* were used to construct the phylogenetic tree.

**Figure 2 ijms-24-00906-f002:**
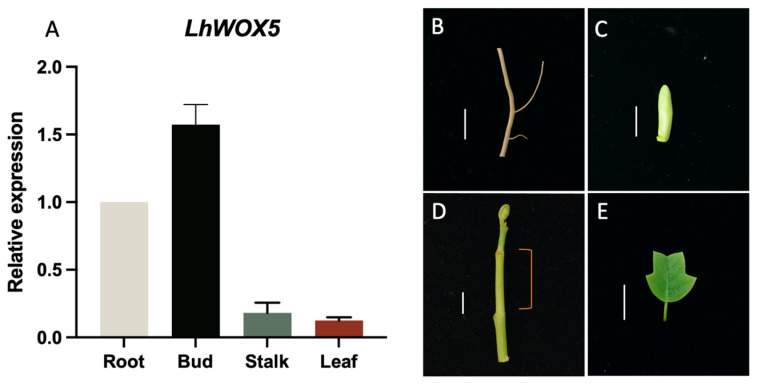
Relative expression of *LhWOX5* in *Liriodendron* hybrids. (**A**) Relative expression levels of *LhWOX5* in different tissues. (**B**–**E**) Different tissues were collected from 3-month-old plants for qRT-PCR, indicated as (**B**) root, (**C**) bud, (**D**) stalk (only the orange area was used for sampling), and (**E**) leaf. Scale bars = 1 cm.

**Figure 3 ijms-24-00906-f003:**
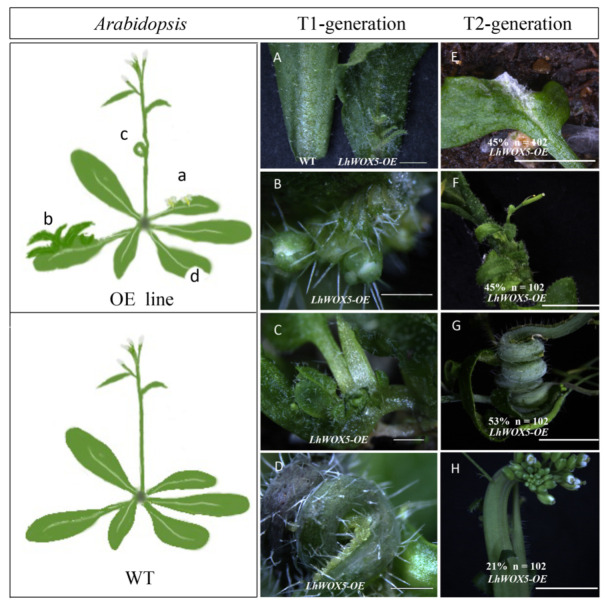
The phenotypes of *p35S:LhWOX5* (*LhWOX5-OE)* transgenic *Arabidopsis*. Hand-drawn images showing transgenic (OE line) and wild type (WT) plants. (a) Ectopic flower buds growing on rosette leaf veins. (b) Ectopic leaves growing on rosette leaf veins. (c) Curly stalk. (d) Incomplete cotyledon. (**A**–**D**) Phenotypes of T1-generation plants. (**A**) rosette leaves of WT and *LhWOX5-OE* lines. (**B**,**C**) Ectopic leaves and/or flower buds growing from rosette leaf veins. (**D**) A curly stalk with undetermined hyperplasia at the curling site. (**E**–**H**) Phenotypes of T2 generation plants. (**E**) Incomplete cotyledon. (**F**) A rosette leaf with ectopic leaves, flower buds, and pods growing on its leaf veins. (**G**) Curly stalk with undetermined hyperplasia. (**H**) Fused stalk. Scale bar = 0.5 um.

**Figure 4 ijms-24-00906-f004:**
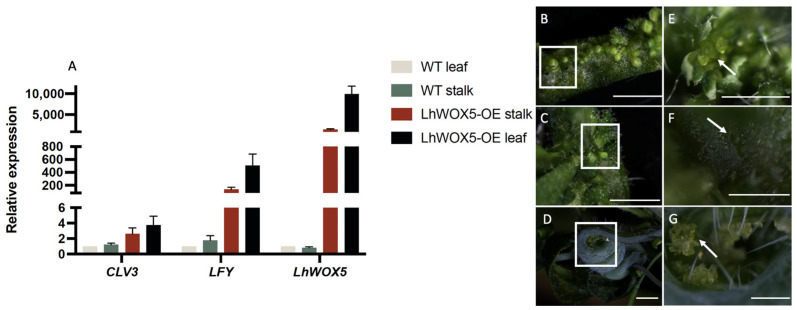
Relative expression of shoot meristem identity genes in *Arabidopsis* (T2 generation) (**A**). (**B**) Shows rosette leaves with ectopic flower buds and leaves. (**C**) Shows the same tissue as (**B**), focusing on the presence of organ primordia on rosette leaves. (**D**) Curly stalk with undetermined hyperplasia. (**E**–**G**) Show enlarged images of square in (**B**–**D**). White arrow indicates opened-flower (**E**), organ primordia (**F**), and undetermined hyperplasia (**G**), respectively. Scale bars: (**B**–**D**), 0.2 um; (**E**–**G**), 0.05 um.

**Figure 5 ijms-24-00906-f005:**
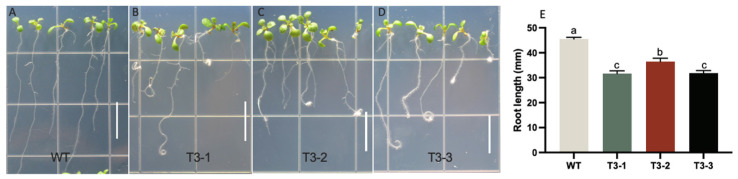
Root phenotypes of *Arabidopsis p35S: LhWOX5*. Analysis of four individual lines, using 8-day-old seedlings for quantification. (**A**) WT, wild type (n = 162). (**B**) Line T3-1 (n = 185). (**C**) Line T3-2 (n = 172). (**D**) Line T3-3 (n = 199). Scale bar = 1 cm. (**E**) Root length in mm, the average ± SE is shown. An ANOVA test was used for statistical analysis. The letters a, b, and c indicate significant differences, with groups marked by identical letters having no significant difference between them.

**Figure 6 ijms-24-00906-f006:**
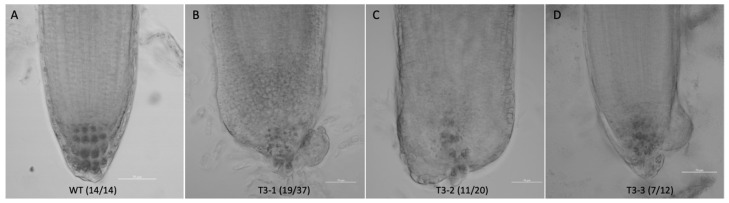
Root morphology of *Arabidopsis p35S: LhWOX5*. The roots of 5–7 days seedlings were analyzed. (**A**) WT, wild type. (**B**) Line T3-1. (**C**) Line T3-2. (**D**) Line T3-3. Scale bar = 50 um, dark gray granules in the root tip indicate starch grains.

**Table 1 ijms-24-00906-t001:** Shoot architecture phenotypes of *p35S:LhWOX5 Arabidopsis*.

Line	Branches	Inflorescences	Leaves	Height (cm)	Number
WT	2.5 ± 0.9	9.38 ± 3.1	20.3 ± 3.6	40.8 ± 1.3	30
T2-1	3.28 ± 1.9 *	9.15 ± 4.7	20.36 ± 5.0	42.9 ± 1.2	34
T2-2	3.19 ± 0.9	12.03 ± 2.6 *	18.83 ± 3.2	36.9 ± 1.4	30
T2-3	2.61 ± 1.4	8.42 ± 4.3	20.69 ± 3.4	43.6 ± 1.4	39

The number of *Arabidopsis* branches, inflorescences, and rosette leaves were counted one month after seedlings were planted in the soil. Height data were collected when plants stopped growing (seedlings were planted in the soil for 42 days). An ANOVA test was used for statistical analysis. *p* < 0.05 (*).

**Table 2 ijms-24-00906-t002:** The analysis of curly roots in *p35S:LhWOX5*.

Line	Percentage of Curly Roots	Number
WT	1.22%	246
Line 1	51.69%	267
Line 2	33.07%	254
Line 3	61.29%	279

## Data Availability

All data generated or analyzed during this study are available within the article or upon request from the corresponding author.

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
