# Peer review of "Overexpression of Liriodenron WOX5 in Arabidopsis Leads to Ectopic Flower Formation and Altered Root Morphology"

_ijms, 2023, doi:10.3390/ijms24020906_

Round 1
Reviewer 1 Report
The manuscript reported a study of the functions of Wuschel-related homeobox 5 gene from Liriodendron, by introducing the gene to Arabidopsis. The authors found that the gene was functional and affected not only root morphology, but also above-ground organs like leaf and floral meristem. The results can be of significant interest to others who work on Liriodendron or other closely related plants. Generally, the manuscript is well-written and provided sufficient data to back up the conclusion of the paper. However, there are some improvements that can be made to the paper to make it more polished and interesting for publication:
1. The introduction can include a bit more justification for why the target gene and target species were chosen in this study, such as by highlighting expected potential uses of the target gene in Liriodendron, or successful application in other crops, etc.
2. I don't have access to Supplementary figures and tables, but I expect that Table S2 should be similar to Table 1, with statistical significance calculation and number of observations included in the table, so that statements in line 118-119 do not look like anecdotal observation.
3. Other than the ability to grow on Kanamycin, were there other steps performed to ensure that all observed T1 and T2 plants were real transgenic plants?
4. Pictures in Figure 3 can be better. As someone unfamiliar with Arabidopsis anatomy, it was quite difficult to understand at first glance. Figure 3 A, B, and D are cropped too closely. If possible, they probably can be zoomed out a little, to make it clearer on which part of the plant the rectangles are located. Brightness, contrast, and clarity are rather poor in all pictures. In the future, please pay more attention to those when taking the photographs, by taking multiple photos while adjusting various settings and lighting. Scale bars seems wrong too. If possible, make them all the same value and vary the bar length instead.
5. The discussion will be more interesting if the applied side of the introduced gene is also deliberated. For example, in the cited paper on poplar, the gene was investigated because it has a possible use in improving the rooting process of stem cuttings. The fact that the Liriodendron gene not only affects roots in unexpected ways but also induces ectopic flower buds clearly will have a significant ramification for its applications. I think such discussion will provide valuable insights for those exploring the use of this gene for such purpose in Liriodendron or other woody plants.
Reviewer 2 Report
The authors reported an overexpression of the WOX5 gene from Liriodendron hybrids (LhWOX5) in Arabidopsis. They found the overexpression led to significant morphological changes in both root and shoot in a woody species. The paper is well written. One of the biggest issues is the lack of novelty. Many researches concentrated on WOX5, and what is the new point here? Just for a trial in woody species like Liriodendron? It may be of interest concerning about its significance in adaptation like drought resistance. However, this point seems lack in this version, which comes to the following question: Why did the authors concentrate on Liriodendron? The current problem that needs to be solved should be clearly depicted in the part of introduction. Also, several concerns below need to be fixed before further revision.
1. AtWOX5 can affect the asymmetric distribution of auxin and thus affect the gravity response, so I wonder if LhWOX5 has the same function?
2. What is the theoretical significance or practical application value of the heterologous expression of LhWOX5 in Arabidopsis thaliana or the species? As the authors pointed out a potential role of LhWOX5 in the development of both the shoot and root, is this significant? In my mind, to change the shoot-root ratio might be of use in practice.
3. Logic and grammar need to be further polished. I only listed some of them.
In the author list, the last word “And” seemed to be in a superscript form
2.1. . Identification and tissue-specific expression pattern analysis of LhWOX5. Two dots here
In results, Figure 2E appeared firstly rather than 2A
Round 2
Reviewer 2 Report
The authors addressed most of my questions. However, the past tense is usually applied when describing the results. And it is not the case in the abstract, I found. So I suggest the authors go through the whole text to revise it.